# FMRP regulates STAT3 mRNA localization to cellular protrusions and local translation to promote hepatocellular carcinoma metastasis

Zhifa Shen [1,2,4,5✉], Bowen Liu[3,4], Biting Wu[1], Hongyin Zhou[1], Xiangyun Wang[3], Jinling Cao[3], Min Jiang[1], Yingying Zhou[1], Feixia Guo[1], Chang Xue [2] & Zai-Sheng Wu[2,5✉]

Most hepatocellular carcinoma (HCC)-associated mortalities are related to the metastasis of cancer cells. The localization of mRNAs and their products to cell protrusions has been reported to play a crucial role in the metastasis. Our previous findings demonstrated that STAT3 mRNA accumulated in the protrusions of metastatic HCC cells. However, the underlying mechanism and functional significance of this localization of STAT3 mRNA has remained unexplored. Here we show that fragile X mental retardation protein (FMRP) modulates the localization and translation of STAT3 mRNA, accelerating HCC metastasis. The results of molecular analyses reveal that the 3′UTR of STAT3 mRNA is responsible for the localization of STAT3 mRNA to cell protrusions. FMRP is able to interact with the 3′UTR of STAT3 mRNA and facilitates its localization to protrusions. Importantly, FMRP could promote the IL-6-mediated translation of STAT3, and serine 114 of FMRP is identified as a potential phosphorylation site required for IL-6-mediated STAT3 translation. Furthermore, FMRP is highly expressed in HCC tissues and FMRP knockdown efficiently suppresses HCC metastasis in vitro and in vivo. Collectively, our findings provide further insights into the mechanism of HCC metastasis associated with the regulation of STAT3 mRNA localization and translation.

[1] Key Laboratory of Laboratory Medicine, Ministry of Education of China, and Zhejiang Provincial Key Laboratory of Medical Genetics, School of Laboratory Medicine and Life Sciences, Wenzhou Medical University, Wenzhou, China. [2] Cancer Metastasis Alert and Prevention Center, Fujian Provincial Key Laboratory of Cancer Metastasis Chemoprevention and Chemotherapy, National & Local Joint Biomedical Engineering Research Center on Photodynamic Technologies, State Key Laboratory of Photocatalysis on Energy and Environment, College of Chemistry, Fuzhou University, Fuzhou, China. [3] Research Center for Molecular Oncology and Functional Nucleic Acids, School of Laboratory Medicine, Xinxiang Medical University, Xinxiang, Henan, China. [4]These authors contributed equally: Zhifa Shen, Bowen Liu. [5]These authors jointly supervised this work: Zhifa Shen, Zai-Sheng Wu. ✉email: shenzhifa@wmu.edu.cn; wuzaisheng@163.com

Hepatocellular carcinoma (HCC) is one of the most common malignant cancers[1]. According to global cancer epidemiology statistics, there are 782,500 newly diagnosed HCC cases worldwide each year, ranking the sixth among all cancer incidences[2]. In clinical settings, primary HCC is generally treated by surgical resection, liver transplantation, and radio-frequency ablation[3]. However, the occurrence of postoperative recurrence and metastasis has led to the death of most HCC patients[4]. Therefore, there is an urgent need to elucidate the molecular mechanism of HCC metastasis and to identify potential targets for the precise treatment of HCC.

Cancer cell motility has a crucial role in the process of tumor metastasis[5]. Cellular motility requires the formation of protrusions at the leading edge of the cells[6,7]. The formation and stability of cell protrusions are closely associated with the ability of cells to migrate and invade, which are the premise of tumor metastasis. Cellular protrusions have been shown to be highly polarized structures that are enriched in many cytoskeletal proteins, adhesion proteins, signaling factors, and other proteins that are responsible for the establishment and maintenance of cellular polarity and directed migration[8,9]. Interestingly, recent studies have discovered that specific mRNAs are enriched in the protrusions of metastatic human cancer cell lines[10–13]. The localization of mRNAs relies on the presence of a cis-acting element, or a zipcode within the mRNA sequence, which is recognized by an RNA-binding protein[14]. Most of these mRNAs encode cytoskeletal proteins and signaling factors located in subcellular compartments and have an important role in the accumulation of the corresponding protein products at the protrusions[15]. However, although many mRNAs encoding transcription factors and nuclear proteins are present in cell protrusions[11,16], the functional significance of this phenomenon in HCC metastasis remains unclear.

Fragile X mental retardation protein (FMRP) is an RNA-binding protein, which modulates the localization, stability, and translation of its target mRNA[17–20]. Until now, the function of FMRP has been deeply explored in the central nervous system. FMRP is encoded by the fmr1 gene[21], and the silence of fmr1 gene will leads to fragile X syndrome (FXS). The FXS is the most common form of inherited intellectual disability, affecting 1:6000 newborns[22,23]. In recent years, several studies have revealed the role of FMRP in cancer progression. The high expression level of FMRP is closely associated with the metastasis of breast cancer[24]. Marine's group has discovered that FMRP acts as an oncogene to promote the invasiveness phenotype in melanoma[25]. However, the association between FMRP and HCC metastasis still needs further exploration.

STAT3 is a well-studied oncogene that has a crucial role in the development of many cancers, including breast, lung, bladder, and colorectal cancers[26–29]. STAT3 is involved in multiple oncogenic processes, such as cell survival, proliferation, angiogenesis, and metastasis, by modulating the expression of various genes[30,31]. In a previous study, we discovered that STAT3 mRNA can specifically localize to the protrusions of metastatic HCC cell lines[32]. However, the underlying mechanism of STAT3 mRNA localization and the functional importance of this localization in HCC metastasis have remained unknown.

In the present study, our results uncovered that FMRP is responsible for the localization of STAT3 mRNA to cell protrusions by interacting with the 3′UTR of STAT3 mRNA, which then promotes the IL-6-mediated translation of STAT3, facilitating HCC metastasis. Our finding reveals a mechanism by which FMRP modulates the localization and translation of STAT3 mRNA and accelerates HCC metastasis, providing potential therapeutic targets for HCC.

## Results

**Observation of STAT3 mRNA localization in HCCLM3 protrusions**. The results of our previous study demonstrated that a variety of transcripts localize to the protrusions of metastatic HCC cells[32]. Direct RNA sequencing (DRS) results showed that STAT3 mRNA was remarkably enriched in the cell protrusion fraction[32]. Thus, to confirm the localization of STAT3 mRNA to cell protrusions, we isolated the cell protrusions of HCCLM3 cells, a characteristic metastatic HCC cell line, and then evaluated the expression of different STAT3 isoforms (Fig. 1a, b and Supplementary Fig. 1a). Two STAT3 isoforms, STAT3α and STAT3β, have a similar molecular structure but differ in the C-terminal region, with STAT3α being the full-length isoform[33,34]. The RT-PCR results showed that STAT3α, rather than STAT3β, was primarily expressed in the cell protrusions (Fig. 1b). Importantly, fluorescence in situ hybridization (FISH) results revealed that STAT3 mRNA was specifically localized in the protrusions of HCCLM3 cells (Fig. 1c). Subsequently, we examined the level of STAT3 expression in clinical HCC tissues. Immunohistochemical (IHC) staining results showed that STAT3 was highly expressed in carcinoma tissues compared with that observed in the non-carcinoma tissues (Fig. 1d, e). Moreover, an analysis of the GEO dataset (GSE28248)[35] showed that STAT3 expression in metastatic HCC tissues was markedly higher than that observed in non-metastatic HCC tissues (Fig. 1f). To further validate the effect of STAT3 on HCC metastasis, we constructed two cell lines in which STAT3 was stably silenced, and the interference efficiency was examined by Q-PCR and western blot assays (Fig. 1g). Through Matrigel Transwell invasion and wound healing migration assays, we confirmed that STAT3 depletion notably inhibited the metastasis of HCCLM3 cells (Fig. 1h–j). Collectively, these results indicated that STAT3 has a crucial role in HCC cell metastasis. However, whether the localization of STAT3 mRNA to cell protrusions is required for the pro-metastasis function of STAT3 remains unknown.

**The 3′UTR of STAT3 mRNA is associated with its localization to cell protrusions**. To elucidate the underlying mechanism associated with the localization of STAT3 mRNA to cell protrusions, and identify the key region responsible for the localization of STAT3 mRNA, we constructed three fragments based on STAT3α mRNA by deleting different functional domains (Fig. 2a). Through FISH analysis, we observed that exogenous overexpression of STAT3-Δ1 fragment failed to promote localization of STAT3 mRNA to cell protrusions, whereas the overexpression of STAT3 overall, STAT3-Δ2 and STAT3-Δ3 fragment obviously promote this phenomenon (Fig. 2b, c). These data suggested that the 3′UTR may have a crucial role in the localization of STAT3 mRNA to cell protrusions. To further assess the localization-promoting element in the 3′UTR of STAT3 mRNA, we constructed six short fragments of the STAT3 3′UTR and separately inserted them into the GFP-MS2 system (Fig. 2d, e). After transfection of the above GFP-MS2 plasmids harboring the STAT3 3′UTR constructs into HCCLM3 cells, we performed real-time tracking of RNA in live cells by immunofluorescence (IF) assay. In this system, the GFP protein cannot localize in the cytoplasm unless the inserted RNA fragment contains the cytoplasmic localization element. Interestingly, the STAT3 3′UTR and STAT3 3′UTR-6 inserts were able to induce the cytoplasmic GFP signal, whereas the other five inserts could not (Fig. 2f). Because the secondary structure of RNA has been reported to be of great importance in RNA localization[36], we analyzed the structure of the STAT3 3′UTR-6 sequence with RNA structure software. The results showed the presence of two loops (termed Loop-1 and

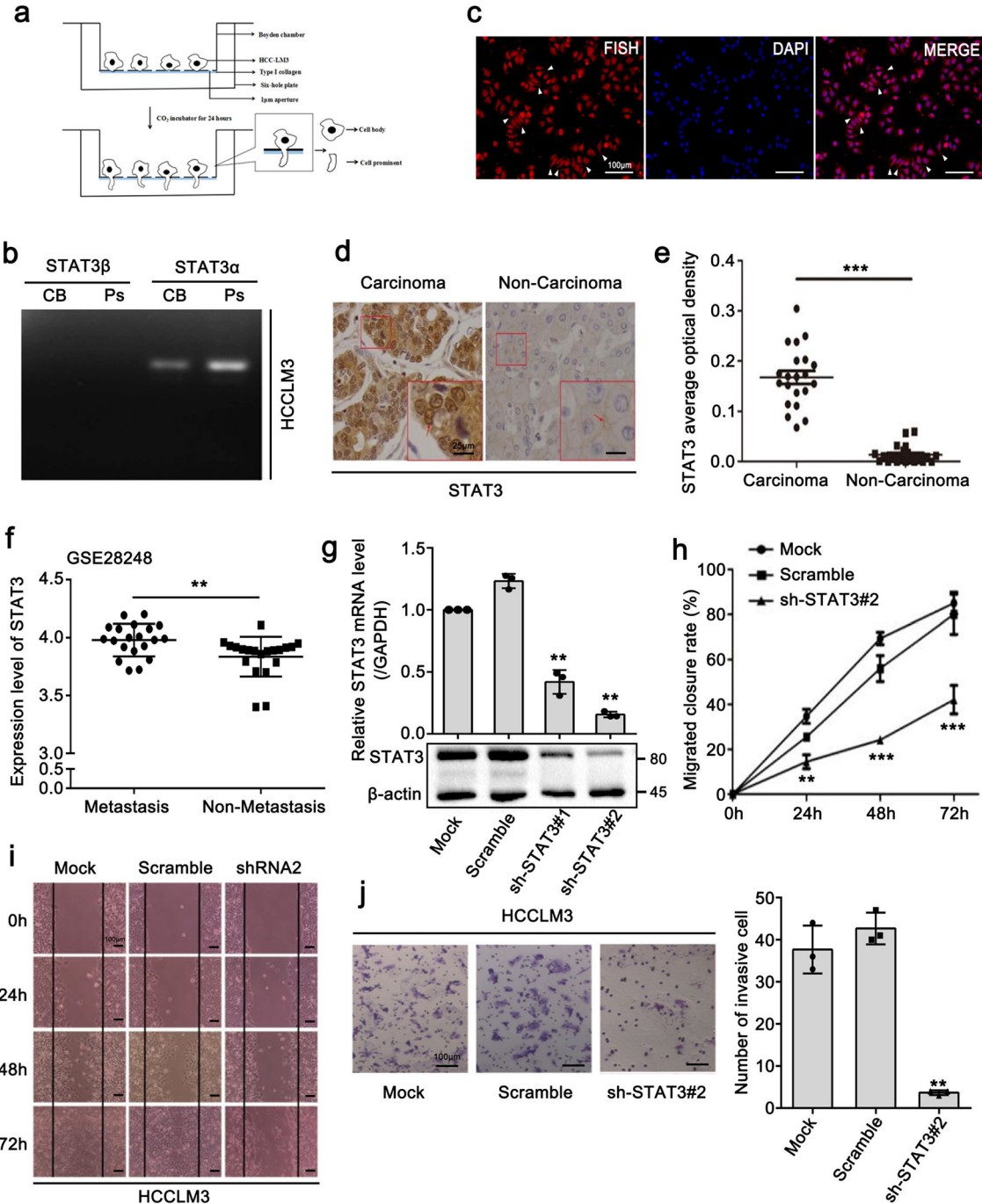

**Fig. 1 Observation of STAT3 mRNA localization in HCCLM3 protrusions. a** Schematic diagram of Boyden chamber. Cells were placed in the upper chamber with a 1-μm pore size microporous membrane, only allowing the migration of cells prominent to the lower chamber. **b** RT-PCR analysis shows that STAT3α, rather than STAT3β, is primarily expressed in the cell protrusions **c** Fluorescence in situ hybridization (FISH) detection to indicate the localization of STAT3 mRNA in HCCLM3 cells. Scale bar: 100 μm. **d** Representative images of STAT3 by immunohistochemical (IHC) staining showing high STAT3 expression in carcinoma tissues (left panel) versus non-carcinoma tissues (right panel) (scale bar: 25 μm (in the enlarged image), $n = 21$ biologically independent samples). Interestingly, STAT3 was mainly gathered in the nucleus (indicated by red arrow, left panel) in carcinoma tissues whereas scattered on the edge of the cell membrane in non-carcinoma tissues (indicated by red arrow, right panel). **e** Statistics of average optical density of STAT3 in **d**. **f** Analysis of STAT3 expression in 20 metastatic HCC tissues versus counterpart non-metastatic tissues in GEO dataset (GSE28248). **g** Stable knockdown of STAT3 in HCCLM3 cells by lentiviral shRNA sequences (shSTAT3). The knockdown effect was verified at both the mRNA and protein levels. **h–j** STAT3 depletion obviously inhibited the migration (**h**, **i**) and invasion (**j**) of HCCLM3 cells. Scale bar: 100 μm. The values in the graphs represent the mean of three biologically independent experiments. Error bars represent ±s.d. *$P < 0.05$, **$P < 0.01$, ***$P < 0.001$ by two-tailed Student's t-test.

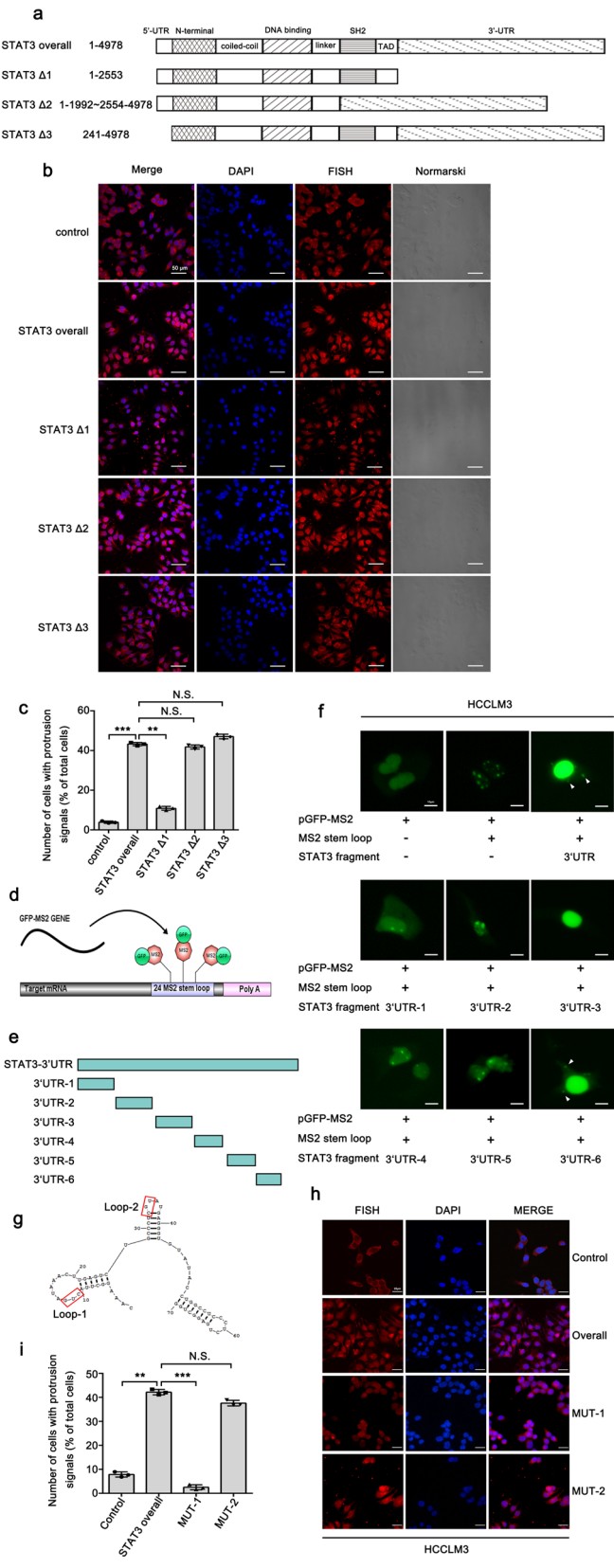

**Fig. 2 The 3′UTR of STAT3 mRNA is associated with its localization to cell protrusions. a** Three fragments were constructed based on STAT3α full-length mRNA by deleting different functional domains. The length of fragments (1–4978) was numbered relative to the first nucleotide of the 5′-UTR. **b** FISH analysis images of STAT3 mRNA expression in control and STAT3 fragments-deleting cells, indicating that overexpression of STAT3 overall, STAT3 Δ2 and STAT3 Δ3 fragment could promote the protrusion-localization of STAT3 mRNA. Scale bar: 50 μm. **c** Quantitative analysis of **b**. Number of cells with protrusion signals reported as a percentage of total cells. **d** Schematic diagram showing the principle of the GFP-MS2 system set-up for detecting cytoplasmic localization element. **e** Six short fragments (3′UTR-1, 3′UTR-2, 3′UTR-3, 3′UTR-4, 3′UTR-5, 3′UTR-6) were constructed from STAT3 3′UTR and separately inserted into the GFP-MS2 system. **f** Immunofluorescence analysis of the GFP-MS2 system inserted with six short fragments from STAT3 3′UTR. STAT3 3′UTR and STAT3 3′UTR-6 can induce the cytoplasm GFP signal in the cell protrusions (indicated by arrow). Scale bar: 10 μm. **g** The secondary structure of STAT3 3′UTR-6 analyzed by RNA structure software showing the potential association with the localization of STAT3 mRNA. **h** Nucleotide substitution mutants were constructed in two different sites of STAT3 3′UTR-6 (mutation of loop1 termed as MUT-1, mutation of loop2 termed as MUT-2). The effect of site mutations on the protrusion-localization of STAT3 mRNA was evaluated by FISH assay. Scale bar: 25 μm. **i** Quantitative analysis of **h**. Number of cells with protrusion signals reported as a percentage of total cells. The values in the graphs represent the mean of three biologically independent experiments. Error bars represent ±s.d. *$P < 0.05$, **$P < 0.01$, ***$P < 0.001$ by two-tailed Student's $t$-test.

protrusions by FISH analysis (Fig. 2h, i). STAT3 mRNA could not localize to the cell protrusions when the indicated site of Loop-1 was mutated (Fig. 2h, i), suggesting that Loop-1 in the 3′UTR-6 region of STAT3 has an essential role in the localization of STAT3 mRNA to cell protrusions.

**FMRP interacts with STAT3 mRNA and facilitates its localization to cell protrusions.** FMRP has been reported to interact with STAT3 mRNA and participate in its localization in axons. Therefore, we assessed whether FMRP is involved in the localization of STAT3 mRNA to HCC cell protrusions. The eukaryotic expression plasmid (pCMV) was used to express Flag-tagged FMRP in HCCLM3 cells. RNA immunoprecipitation (RIP) assay results showed that exogenous FMRP interacted with STAT3 mRNA (Fig. 3a). Meanwhile, we performed a RIP assay with an FMRP antibody in HCCLM3 cells, and the results revealed an interaction between endogenous FMRP and STAT3 mRNA (Fig. 3b). In addition, RIP assay in isolated cell protrusions also uncovered this interaction (Fig. 3c). Similarly, the confocal microscopy observations demonstrated that FMRP mainly colocalized with STAT3 mRNA in the cytoplasm and protrusions of HCCLM3 cells (Fig. 3d, e). To determine whether FMRP directly binds to the 3′UTR of STAT3 mRNA, a 500 bp sequence of the STAT3 mRNA 3′UTR starting from its stop codon was used in RNA gel-mobility shift assays. As shown in Fig. 3f, FMRP directly interacted with the radiolabeled STAT3 3′UTR. However, this interaction was remarkably disrupted upon the addition of unlabeled STAT3 3′UTR but not unlabeled non-specific RNA (Fig. 3f). Furthermore, we also performed the gel-mobility shift assay by adding the unlabeled STAT3 3′UTR MUT-1 and MUT-2. The data showed that the addition of unlabeled STAT3 3′UTR MUT-1 was unable to abolish the interaction of FMRP with radiolabeled STAT3 3′UTR, while the addition of MUT-2 still could abolish it, suggesting that the loop1 of STAT3 3′UTR was responsible for the interaction between FMRP and STAT3 3′UTR (Fig. 3g). To further investigate the role of FMRP in the

Loop-2) in the secondary structure of STAT3 3′UTR-6 that may be associated with the localization of STAT3 mRNA to cell protrusions (Fig. 2g). Then, we constructed nucleotide substitution mutants in the indicated sites within the two loops (MUT-1: site-mutation of Loop-1; MUT-2: site-mutation of Loop-2) and evaluated their effects on the localization of STAT3 mRNA in cell

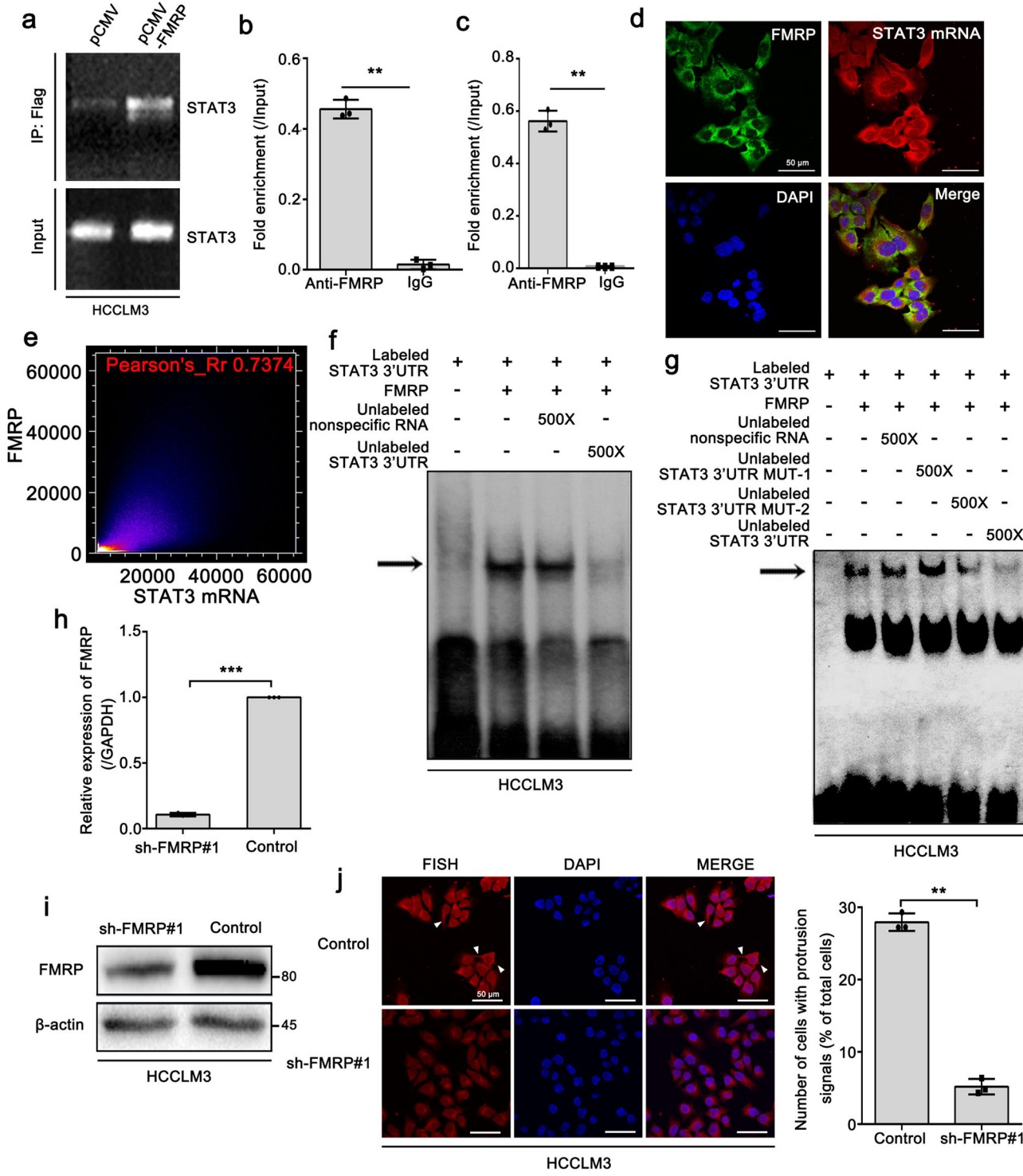

localization of STAT3 mRNA to cell protrusions, we used a shRNA to specifically knockdown FMRP in HCCLM3 cells (Fig. 3h, i). Strikingly, FMRP knockdown markedly abolished the localization of STAT3 mRNA to cell protrusions (Fig. 3j), indicating that FMRP is required for the localization of STAT3 mRNA in HCCLM3 protrusions. Taken together, these results indicate that FMRP interacts with STAT3 mRNA and facilitates its localization to cell protrusions.

**FMRP promotes the IL-6-mediated translation of STAT3 mRNA.** The subcellular localization of mRNA is an evolutionarily

conserved process[37,38] that has been reported to be closely associated with its subsequent translation[39]. To elucidate the role of the FMRP-mediated localization of STAT3 mRNA to cell protrusions, we used IL-6 as a STAT3 activator to assess whether FMRP is involved in the translation of STAT3 mRNA. After being activated by IL-6, STAT3 is phosphorylated by the receptor-associated kinase JAK and then forms homo or heterodimers that translocate into the cell nucleus where it functions as a transcription activator[40]. The Q-PCR and Western blot results showed that the protein level of STAT3 and p-Tyr705-STAT3 was increased after IL-6 treatment, while that of STAT3 mRNA was unaffected (Fig. 4a and Supplementary Fig. 2a). These

**Fig. 3 FMRP interacts with STAT3 mRNA and facilitates its localization to cell protrusions. a** RNA immunoprecipitation (RIP) assay showing the binding of exogenous FLAG-tagged FMRP and STAT3 mRNA in HCCLM3 cells. **b** The interaction of endogenous FMRP with STAT3 mRNA in HCCLM3 cells assessed by RIP-qPCR. Error bars represent ±s.d. ***$P < 0.001$ (IgG compared with anti-FMRP) by two-tailed Student's $t$-test. **c** The interaction of endogenous FMRP with STAT3 mRNA in the protrusions of HCCLM3 cells assessed by RIP-qPCR. Error bars represent ± s. d. ***$P < 0.001$ (IgG compared with anti-FMRP) by two-tailed Student's $t$-test. **d** FISH assay displaying the colocalization of FMRP and STAT3 mRNA in the cytoplasm and protrusions of HCCLM3 cells. Scale bar: 50 μm. **e** Quantitative analysis of colocalization of **d**. The colocalization image was converted into a visual scatter plot. Most of the points are distributed on the diagonal, and the Pearson coefficient is 0.7374. In this case, it indicated that FMRP and STAT3 mRNA are colocalized. **f** FMRP binds to the 3′UTR of STAT3 mRNA. Aliquots of $^{32}$P-labeled 3′ UTR of STAT3 mRNA were incubated with recombinant FMRP. RNA-protein complexes (indicated by arrow) were formed when the RNA probe incubated with recombinant FMRP. The complexes were competed by 500× excess of unlabeled 3′ UTR of STAT3 mRNA, but not by the non-specific RNA (random tRNA). **g** The complexes of $^{32}$P-labeled 3′ UTR of STAT3 mRNA with FMRP were competed by 500× excess of unlabeled STAT3 3′ UTR MUT-2, but not by the unlabeled STAT3 3′ UTR MUT-1 and non-specific RNA (Random tRNA). **h**, **i** Stable knockdown of FMRP in HCCLM3 cells by lentiviral shRNA sequences (shFMRP). The knockdown effect was verified at both the mRNA and protein levels. **j** FISH imaging of STAT3 mRNA showing that knockdown of FMRP markedly abolished the protrusion-localization of STAT3 mRNA. The right panel is the quantitative analysis. Number of cells with protrusion signals was reported as a percentage of total cells. Scale bar: 50 μm. The values in the graphs represent the mean of three biologically independent experiments. Error bars represent ±s.d. *$P < 0.05$, **$P < 0.01$, ***$P < 0.001$ by two-tailed Student's $t$-test.

data suggest that IL-6 might regulate the expression of STAT3 at the translational level in HCCLM3 cells. In addition, we observed that the level of nuclear phosphorylated STAT3 was upregulated after IL-6 treatment for 6 h (Fig. 4b). Similarly, IF assay results uncovered that the time-dependent treatment of IL-6 notably enhanced the expression and subsequent nuclear importation of STAT3 protein (Fig. 4c). The findings were consistent with the aforementioned study[40] and indicated the IL-6-mediated modulation of STAT3 protein expression. Thus, we evaluated whether FMRP can affect STAT3 expression in an IL-6-dependent manner. We constructed two cell lines with stably silenced FMRP and assessed the expression level of STAT3 in these cells. Compared with that observed in the control cells, the protein level of STAT3 was downregulated in the FMRP-knockdown cells, whereas STAT3 mRNA levels were not notably altered (Fig. 4d), indicating that FMRP may also modulate STAT3 expression at the translational level. Furthermore, FMRP overexpression (OE-FMRP) enhanced the expression and nuclear importation of STAT3 (Fig. 4e). Interestingly, the increase in STAT3 protein levels mediated by IL-6 was abolished after cells were treated with cycloheximide (CHX, a protein synthesis inhibitor) (Fig. 4f), indicating that IL-6 modulates STAT3 expression at the translational level, rather than the post-translational level. However, when FMRP was silenced, IL-6 could not alter STAT3 protein levels in HCCLM3 cells regardless of CHX treatment (Fig. 4f). This result suggests that FMRP is responsible for the IL-6-mediated translation of STAT3. The phosphorylation of FMRP has been reported to be important for its function in cells[41]. In the present study, using the human protein reference database online resource (http://www.hprd.org/), we identified four serine residues as potential phosphorylation sites that may be involved in the function of FMRP in IL-6-mediated STAT3 translation. Therefore, we mutated these serine residues to alanine residues and then separately transferred the FMRP mutants into HCCLM3 cells to evaluate their effect on IL-6-mediated STAT3 translation. Western blot analysis results showed that IL-6 failed to enhance the expression of STAT3 when S114 was mutated (Fig. 4g). Taken together, these results indicate that FMRP can promote the IL-6-mediated translation of STAT3 in HCCLM3 cells.

**FMRP knockdown suppresses the metastatic capacity of HCCLM3 in vitro.** Since FMRP was shown to have a crucial role in STAT3 mRNA localization and translation, we subsequently assessed the correlation between FMRP with HCC metastasis. IHC staining results using clinical HCC tissues revealed that FMRP expression in carcinoma tissues was remarkably higher than that observed in the non-carcinoma tissues (Fig. 5a, b). Next, we evaluated the levels of FMRP expression in 8 HCC and paired non-tumor tissues. The results revealed that FMRP was highly expressed in HCC tissues compared to that observed in the non-tumor tissues (Fig. 5c). Moreover, FMRP expression levels in the highly metastatic cell lines HCCLM3 and MHCC97H were notably higher than the cell line MHCC97L, which has comparatively lower metastasis (Fig. 5d). Next, we performed wound healing migration and Transwell migration/invasion assays to evaluate the metastatic capacity of FMRP-knockdown cells, before which Q-PCR and Western blot analyses were performed to examine the FMRP level in the indicated cells (Fig. 5e, f). Literatures have reported that cell density can affect cell migration by modulating STAT3 activity[42,43]. We evaluated the level of STAT3-pTyr705 in different cell densities and found that STAT3 can indeed be activated (Supplementary Fig. 3a, b). But the further data showed the cell migration was notably suppressed with FMRP knockdown at each cell density (Supplementary Fig. 3c). This result excluded the possibility that FMRP promoted cell migration via cell density-activated STAT3. Strikingly, FMRP knockdown remarkably inhibited the migration and invasion abilities of HCCLM3 cells (Fig. 5g, h). Furthermore, we performed CCK-8, Transwell migration and invasion assays to examine the role of S114 in FMRP with respect to cell proliferation, migration, and invasion. As shown in Supplementary Fig. 4, compared with that observed in the control group, the proliferation rate, migrated and invasive cell number in the FMRP-WT group was markedly increased. However, this increase was abolished when the S114 was mutated. Altogether, these results indicate that FMRP can accelerate HCC cell metastasis and the S114 in FMRP is responsible for FMRP-mediated cell proliferation, migration, and invasion.

**FMRP knockdown suppresses the metastatic capacity of HCCLM3 in vivo.** To further confirm the role of FMRP in HCC metastasis in vivo, we established an in situ human hepatocellular carcinoma model by injecting FMRP-knockdown and control cells into nude mice through in situ hepatic injection. We observed that stable FMRP knockdown decreased the volumes and weights of tumors compared to that observed in the control groups, indicating that stable knockdown of FMRP effectively suppressed the growth of liver tumors (Fig. 6a–c). In addition, Q-PCR and Western blot analysis of liver tumors showed that FMRP knockdown remarkably suppressed the expression of STAT3 protein, whereas STAT3 mRNA level remained unaffected (Fig. 6d, e), further verifying that FMRP modulate STAT3

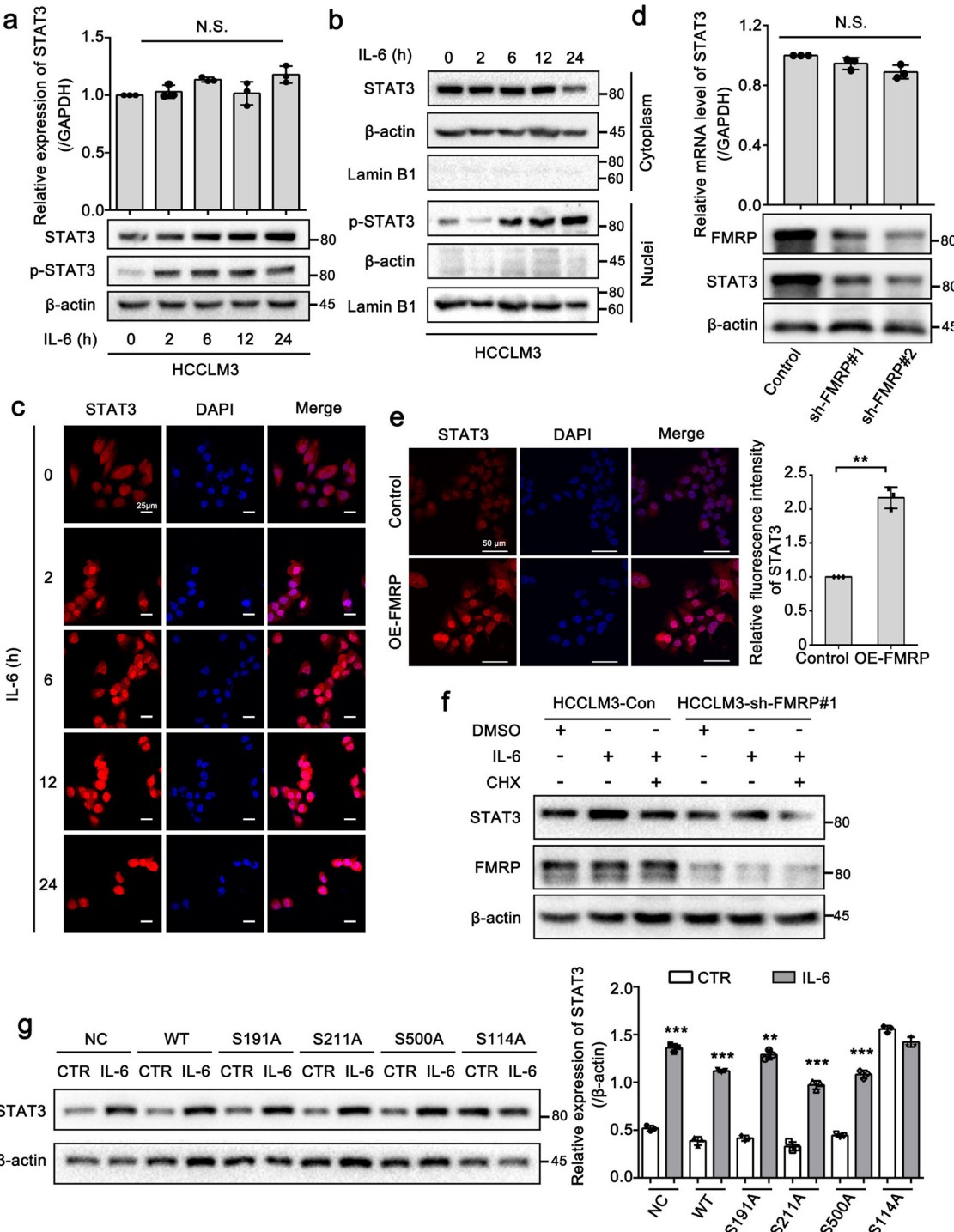

**Fig. 4 FMRP promotes the IL-6-mediated translation of STAT3 mRNA. a** Effect of STAT3 activator IL-6 (50 ng/mL) on the expression of STAT3. Q-PCR and Western blot analysis of the relative expression of STAT3 mRNA and protein after IL-6 treatment in the indicated time points. **b** Western blot of nuclear and cytoplasmic distribution of total STAT3 and pSTAT3 after the treatment of IL-6 in the indicated time points. **c** Immunofluorescence analysis of STAT3 expression after IL-6 treatment in the indicated time points. Scale bar: 25 μm. **d** Q-PCR and Western blot analysis of STAT3 expression in control and two FMRP-knockdown HCCLM3 cells. **e** Immunofluorescence analysis of STAT3 in control and FMRP-overexpressing HCCLM3 cells. Scale bar: 50 μm. **f** Western blot analysis of STAT3 expression in HCCLM3-Control and HCCLM3-sh-FMRP#1 cells treated with 100 μg/mL cycloheximide (CHX) and/or IL-6 (50 ng/mL). **g** Western blot analysis of the effects of wild-type FMRP (WT) and mutated FMRP on the IL-6-mediated STAT3 translation. IL-6 failed to enhance the expression of STAT3 when the S114 was mutated. The right panel is the quantification of the intensity relative to β-actin. The values in the graphs represent the mean of three biologically independent experiments. Error bars represent ±s.d. *$P < 0.05$, **$P < 0.01$, ***$P < 0.001$ by two-tailed Student's $t$-test.

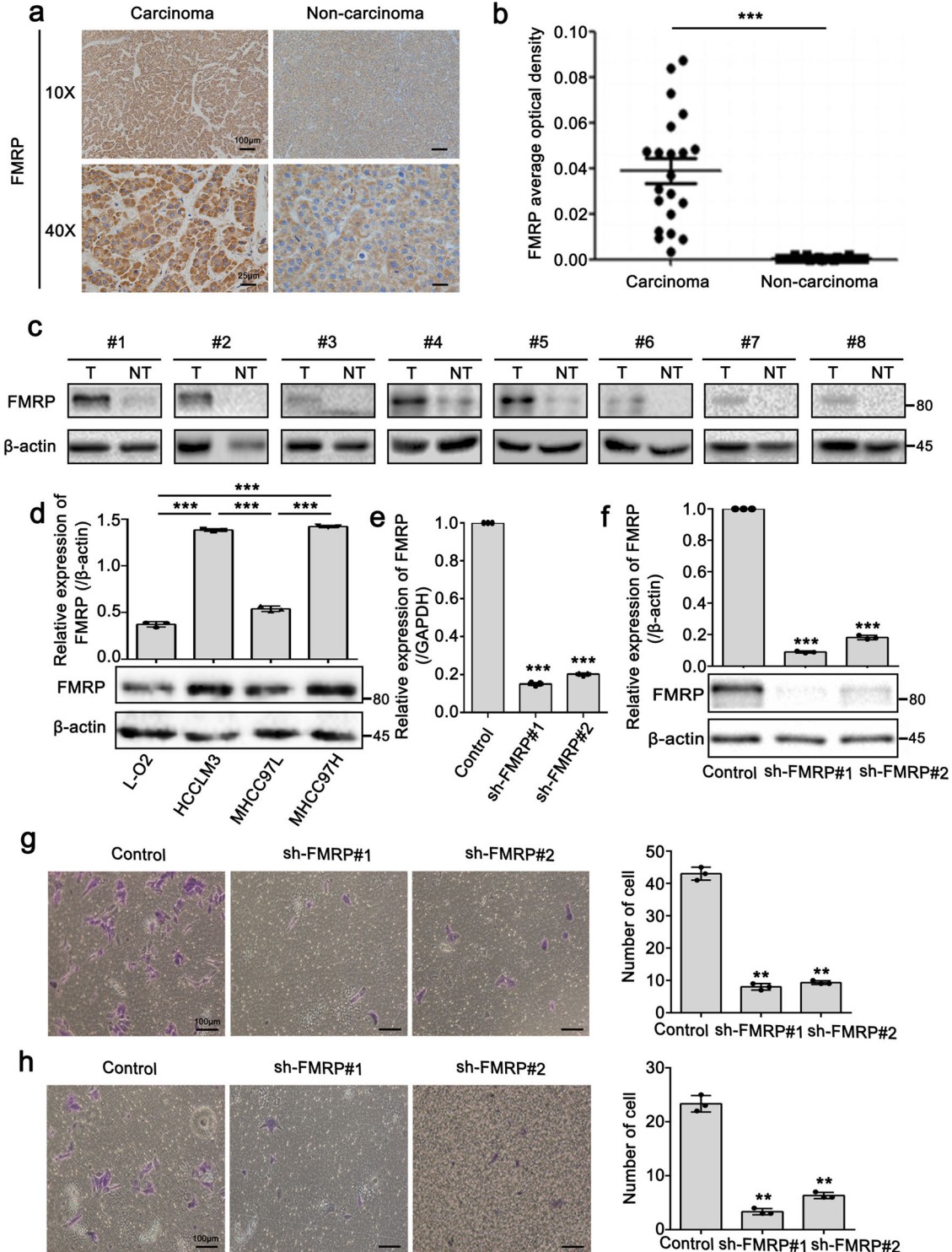

**Fig. 5 FMRP knockdown suppresses the metastatic capacity of HCCLM3 in vitro. a**, **b** Representative IHC staining (**a**) and statistics of average optical density of FMRP (**b**) in matched carcinoma tissues and non-carcinoma tissues. $n = 21$ biologically independent samples. Scale bar (10×): 100 μm. **c** Western blot analysis of FMRP protein levels in 8 paired HCC tissues (T) and adjacent non-tumor tissues (NT). **d** Western blot analysis of FMRP in L-O2, HCCLM3, MHCC97L, and MHCC97H cells (lower panel) and quantification of the intensity relative to β-actin (upper panel). **e**, **f** Q-PCR (**e**) and western blot (**f**) analyses examined the knockdown efficiency of FMRP in FMRP-knockdown cells. **g**, **h** Transwell migration (**g**) and invasion assays (**h**) in Control and FMRP-knockdown cells. The right panel is the quantification of the left panel. Scale bar: 100 μm. The values in the graphs represent the mean of three biologically independent experiments. Error bars represent ±s.d. *$P < 0.05$, **$P < 0.01$, ***$P < 0.001$ by two-tailed Student's $t$-test.

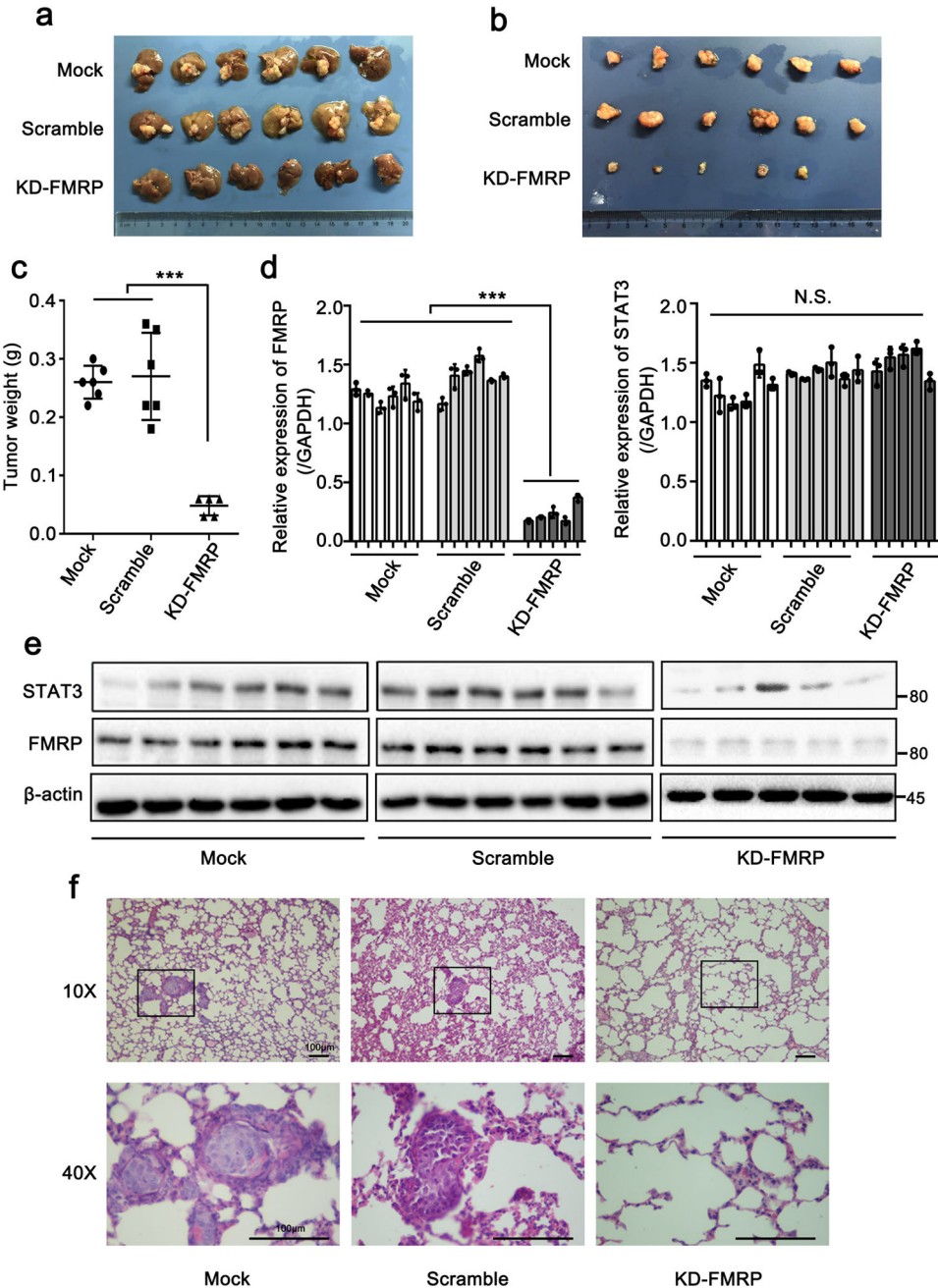

**Fig. 6 FMRP knockdown suppresses the metastatic capacity of HCCLM3 in vivo. a** Livers injected with FMRP-knockdown cells to form xenograft tumors, accompanied by a scramble and mock groups. $n = 6$ biologically independent samples (one mouse in the KD-FMRP group did not develop a tumor). **b** Image of the xenograft tumors stripped from livers in nude mice. **c** Weights of xenograft tumors in three groups were measured. **d**, **e** Q-RCR (**d**) and western blot (**e**) analysis of FMRP and STAT3 expression of the xenograft tumors. The values in the graphs represent the mean of three biologically independent experiments. **f** Representative H&E staining of metastatic lung nodules at the indicated groups. Scale bar: 100 μm. Error bars represent ±s.d. *$P < 0.05$, **$P < 0.01$, ***$P < 0.001$ by two-tailed Student's $t$-test.

expression at the translational level. Next, H&E staining results showed that the FMRP knockdown group was attenuated for lung metastasis (Fig. 6f), suggesting that FMRP contributes to lung metastasis. We also evaluated the role of S114 in FMRP with respect to HCC metastasis in vivo. We observed markedly decreased liver tumor volume and weight in the FMRP-S114A group compared to that measured in the FMRP-WT group (Supplementary Fig. 5a–c). Furthermore, more lung metastasis nodules were observed in the FMRP-WT group than that in the FMRP-S114A group (Supplementary Fig. 5d). Taken together, these results indicate that FMRP contributes to the growth and

metastasis of HCC in vivo and the S114 in FMRP has a crucial role in its function.

## Discussion
Although important achievements have been made in HCC diagnosis and therapy in recent years, the clinical outcome for HCC patients remains unsatisfactory due to untimely diagnosis, relapse, and the high probability of HCC cell invasion and metastasis[44]. The anti-angiogenic drug sorafenib is currently the only systemic treatment for liver cancer, but the median survival

time of sorafenib-treated patients is only one year[45]. Thus, preventing tumor metastasis by controlling HCC cell migration and invasion remains a major challenge. In recent years, studies have shown that protrusions of migrated cells have a crucial role in the process of cell migration and invasion[46,47]. However, the associated components and biological function of cell protrusions in HCC metastasis have remained un-elucidated.

In the present study, we purified and identified the transcripts specifically enriched in the protrusions of highly metastatic HCC cells. As previously reported, a large number of protrusion-enriched mRNAs encoding signaling protein, cytoskeletal and translation factor were observed[13]. Strikingly, we also detected transcripts encoding transcription factors and other nuclear proteins. Among these transcripts, STAT3 mRNA was of particular interest, since it was abundantly accumulated in the cell protrusions. Furthermore, the results of several studies have revealed that STAT3 has a key role in the migration and invasion of HCC cells[30,48]. This finding led us to consider the potential association of STAT3 mRNA localization with the promotion of HCC metastasis. Interestingly, our data revealed the underlying mechanism for the localization of STAT3 mRNA to cell protrusions and demonstrated its functional importance in HCC metastasis.

In the present study, we showed that FMRP promotes the localization of STAT3 mRNA to cell protrusions and its subsequent translation, facilitating HCC metastasis. FMRP is a classic RNA-binding protein involved in different steps of RNA metabolism[49]. Its function involves the trafficking, translation, and translocation of mRNAs[50]. Up to now, only a few RNA-binding proteins, such as ZBP1 and APC, have been shown to be involved in the localization of specific mRNAs at cell protrusions[14]. ZBP1 binds mRNAs encoding cytoskeletal proteins such as β-actin, α-actinin, and Arp2/3complex and promotes their localization at the leading edge of cells[51]. In addition, ZBP1 can interact with a cis-element (or zip code) present in the 3′UTR of β-actin mRNA to promote its proper localization. In the present study, we demonstrated that FMRP was responsible for the localization of STAT3 mRNA to cell protrusions by interacting with its 3′UTR. FMRP is a well-established repressor of translation that functions by blocking the translational initiation or elongation step[52]. However, accumulating evidence has also demonstrated that the effect of FMRP on translational regulation is not exclusively a repressive action. For instance, FMRP has been shown to function as a translational activator for the mRNA encoding Kv4.2[49,53]. Moreover, FMRP also positively regulates the expression of ACSL1 and Sod1 by separately interacting with their mRNAs[54,55]. In our present study, we showed that FMRP promotes the IL-6-mediated translation of STAT3 mRNA, consistent with the results of the aforementioned investigations. Although several studies have demonstrated that Ser500 of FMRP has a crucial role in translational regulation, in the present study, we showed that S114 of FMRP is responsible for its ability to modulate STAT3 mRNA translation. In terms of function, our data revealed that FMRP is highly expressed in HCC tissues and promotes HCC metastasis in vitro and in vivo, clearly demonstrating FMRP can alter HCC progression by modulating STAT3 expression. Therefore, FMRP might be a potential therapeutic target for the precise treatment of HCC metastasis.

In summary, we elucidated a mechanism involved in promoting HCC metastasis. FMRP drives the localization of STAT3 mRNA to protrusions by interacting with the 3′UTR of STAT3 and facilitates IL-6-mediated STAT3 mRNA translation, accelerating HCC metastasis. The results of our present study reveal a molecular basis for treating STAT3-associated HCC metastasis. More importantly, our findings reveal the crucial role of FMRP in cancer progression, providing a potential target for HCC treatment.

## Methods

**Cell culture and transfection**. HCCLM3, L-O2, MHCC97L and MHCC97H cells were obtained from the Chinese Academy of Sciences. All cell lines were cultured in DMEM (Gibco, USA) supplemented with 10% FBS (Gibco) at 37 °C under an atmosphere with 5% $CO_2$. The experimental operation will be started when the cell density reaches above 90%. FMRP knockdown was performed using the plasmid pLKO.1 (Addgene) following the manufacturer's protocol. The sequences of oligonucleotides designed and used in the present study are shown in Supplementary Table 1.

**Immunohistochemical staining**. The levels of FMRP and STAT3 expression were assessed in HCC tissues and matched noncancerous tissues by immunohistochemical analysis. After dewaxing, antigen retrieval was performed by microwaving the samples in sodium citrate buffer (pH 8.0) for 20 min. Subsequently, the slides were incubated in 3% $H_2O_2$ and then blocked by incubation in 10% normal goat serum for 30 min at room temperature. Rabbit anti-STAT3α and mouse anti-FMRP antibodies were purchased from (CST, USA) and (Abcam, UK), respectively. The slides were incubated with the corresponding primary antibody at 4 °C overnight. Then, after being washed with PBS, the slides were incubated with the secondary antibody for 1 h at 37 °C, with diaminobenzidine (DAB) (Beijing Zhongshan Golden Bridge Biotechnology, China) used as the chromogen. All human liver tissue slides were obtained from the First Affiliated Hospital of Wenzhou Medical University. The clinical characteristics of patients from whom liver cancer tissue specimens were collected are presented in Supplementary Table 2.

**Immunofluorescence**. Cells were cultured on acid-treated glass coverslips. Subsequently, the cells were fixed with 4% paraformaldehyde for 30 min, washed three times with phosphate-buffered saline (PBS), and then permeabilized with 0.5% Triton X-100 in PBS for 15 min at room temperature. After being washed with PBS three times, the cells were blocked with 0.5% BSA for 2 h before being washed with PBS. Then, the cells were incubated with primary antibodies, including rabbit anti-STAT3α (CST) and mouse anti-FMRP (Abcam), after which they were incubated with secondary antibodies, including Alexa Fluor 488 goat anti-rabbit IgG (Invitrogen, USA) and Alexa Fluor 594 donkey anti-mouse IgG (Invitrogen) before being stained with DAPI for 5 min. Finally, the cells were analyzed using a confocal microscope (Nikon A1R, Japan). The quantification of immunofluorescence was analyzed by Image J software. For the colocalization analysis, the image calculator tool was used to correct the background of the image. After unifying the background, the colocalization finder tool was used to analyze the Pearson correlation coefficient[56–58].

**Plasmid construction**. Four STAT3 gene fragments with different deletions were amplified from human HCCLM3 cell line cDNA and subcloned into the vectors pSP64 poly (A) and pBluescript IISK. The four fragments were named STAT3-overall, STAT3-Δ1, STAT3-Δ2, and STAT3-Δ3. STAT3-overall was subcloned into pSP64 poly (A) vector, while the others were subcloned into the pBluescript IISK plasmid.

**In vitro transcription**. Transcription templates were prepared from the constructed plasmids. The plasmids were digested with SmaI (NEB, USA), and transcription was performed with SP6 RNA polymerase (NEB) to generate STAT3 mRNA (termed as STAT3-overall). The STAT3-Δ1, STAT3-Δ2, and STAT3-Δ3 constructs were digested with SmaI and transcribed with T7 RNA polymerase (NEB), respectively. We used formaldehyde denaturing gel electrophoresis to determine the size of the four RNAs.

**RNA transfection and fluorescence in situ hybridization (FISH)**. HCCLM3 cells were separately transfected with the four RNA fragments as described under in vitro transcription by using Lipofectamine 3000 (life, USA) for 10 h. For in situ hybridization, we used a Fluorescence In Situ Hybridization Kit (Ribibio, China) following the manufacturer's instructions. Total cytosolic STAT3 mRNA was visualized by FISH using a specific RNA oligonucleotide probe labeled with cy3 (Ribibio, China). All FISH images were taken using a confocal microscope (Nikon A1R, Japan).

**RNA immunoprecipitation (RIP)**. We cultured four dishes (10 cm) of HCCLM3 cells and harvested them with a cell scraper (Costar, USA) in 10 mL of ice-cold PBS, which were then collected by centrifugation and broken up with polysome lysis buffer. Protein A + G agarose (Millipore, USA) was incubated with 5 μl of rabbit anti-FMRP (Abcam, UK) or normal IgG (Beyotime, China) for 1 h. Cell lysates were stored at −80 °C overnight before being centrifuged. Then, the A + G agarose-antibody was incubated with the cell lysates at 4 °C overnight. The following day, the agarose was washed with 1 mL of NT2 buffer (supplemented with 1 M urea, 0.1% SDS, and 150 mM NaCl) six times. NT2 buffer supplemented with 180 μl of proteinase K was used to digest the proteins by incubating the tubes at 55 °C for 30 min with shaking. Then, 1 mL of TRIzol reagent was added for RNA extraction, after which 2 μl of glycogen (20 mg/ml) in isopropanol was added to

precipitate the RNA. To avoid DNA contamination, we used reverse transcription kits (Takara, Japan), which are able to remove DNA.

**Cell invasion and migration assays.** Cell invasion and migration were assessed using Transwell inserts (8 μm pore diameter). The upper chamber was precoated with BD Matrigel. The cells were added to the upper chamber with a serum-free medium ($2.5 \times 10^5$ cells/well), and the resulting chamber was filled with 700 μl of complete cell culture medium containing 18% FBS. After incubating for 48 h, the upper chamber was carefully swabbed with cotton, and then the cells that had invaded through the membrane were fixed with 4% paraformaldehyde and stained with crystal violet for 15 min. Migration ability was assessed using the same procedure with the following modifications: (i) the Matrigel coating was omitted, (ii) the cell density was $2.5 \times 10^5$ cells/well, and (iii) the cells were cultured for 24 h. Finally, the number of invaded and migrated cells were counted using a microscope. Wound healing migration assays were carried out by generating a vertical scratch on a monolayer of HCCLM3 cells. Images were captured under an inverted microscope after 0 h, 24 h, 48 h and 72 h, respectively, and the wound area was calculated in five randomly selected microscopic fields. The experiments were performed three times.

**Xenograft tumor metastasis assay.** Thirty male BALB/C-nu nude mice weighing 18–20 g were provided by the Experimental Animal Center of Wenzhou Medical University. All animal experiments were approved by the Zhejiang Management Committee for Medical Laboratory Animal Sciences, and the corresponding cells were collected and suspended at a density of $2 \times 10^6$ cells/mL in 0.1 mL of 1:1 PBS/ Matrigel (BD Biosciences). The cells in the logarithmic growth phase were centrifuged and resuspended under standard conditions to be injected into the liver into corresponding groups of nude mice (6 mice/group). Seven weeks after injection, the nude mice were killed by cervical dislocation, and the weights of the tumors were determined. In addition, liver and lung tissues were collected and processed for histological and cytological evaluation.

**Cell proliferation.** Cell proliferation rates were measured using a Cell Counting Kit-8 (CCK-8) (Lot: ND657, Dojindo) according to the manufacturer's instructions. Briefly, the cells were plated in 96-well plates at a density of 2000 cells/well in 100 μL of culture medium. The following day, 10 μL of CCK-8 reagent was added to each well, and the cells were then incubated under a humidified atmosphere of 5% $CO_2$ for 2 h at 37 °C. The absorbance of the cell sample was measured at a wavelength of 450 nm using a microplate reader. Cell proliferation assay was assessed at 0, 12, 48, 72, and 96 h, with each sample assayed in five replicate wells, and the experiments were performed three times independently.

**Histological analysis.** The excised lungs were closely examined, after which the lungs were irrigated and immersed in a picric acid fixative for routine treatment. The picric acid-immobilized, paraffin-embedded slides were cut into 5-μm sections, the slides were stained with hematoxylin and eosin (H&E), and intrapulmonary metastasis was observed with a microscope.

**Statistics and reproducibility.** All replicates displayed in this paper are biological replicates; technical replicates (usually three) were performed and used to generate the means for each biological replicate. The sample sizes and number of replicates were indicated in the figure legends. Statistical significance was assessed using GraphPad Prism 8. The values in the graphs represent the mean of three biologically independent experiments. Error bars represent ±s.d. *$P < 0.05$, **$P < 0.01$, ***$P < 0.001$ by two-tailed Student's $t$-test.

**Reporting summary.** Further information on research design is available in the Nature Research Reporting Summary linked to this article.

## Data availability

The source data underlying the graphs presented in the main figures are shown as Supplementary data 1. Uncropped blots of major figures are shown in Supplementary Fig. 6. All other data supporting the findings of the study are available within the paper and Supplementary information. Data relating to Supplementary figures are available from Zhifa Shen or Zai-Sheng Wu upon reasonable request.

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

## Acknowledgements

This work was supported by the Major Science Technology Projects of Wenzhou, Zhejiang, China (NO. 2016Y0852), the Science Technology Projects of Wenzhou, Zhejiang, China (NO. Y20170001), the Science and Technology Projects of Zhejiang, China (Grant NO: LY18H160051), the Major Science and Technology Plans for Medicine and Health of Zhejiang, China (Grant NO: WKJ-ZJ1723) and the Key Project of Natural Science Foundation of Fujian Province (Grant NO: 2019J02005).

## Author contributions

Z.S., B.L., B.W., H.Z., M.J., X.W., J.C., and Y.Z. designed and performed the experiments and interpreted the data. C.X. helped with the image recordings. F.G. carried out the animal experiment. B.L. designed the study, provided valuable advices, and wrote the first draft of the paper. Z.S. and Z.-S.W. conceived the project, interpreted the data, and co-wrote the manuscript. All authors read the paper and made contributions.

## Competing interests

The authors declare no competing interests.
