## [Peer Review File · Communications Biology]

Reviewers' comments:

Reviewer #1 (Remarks to the Author):

The manuscript entitled "Involvement of FMRP in STAT3 mRNA localization to cellular protrusions and local translation promoting HCC metastasis" by Shen Z, et al. presented an interesting study on how FMRP regulated the translation of STAT3 by binding to its mRNA. The authors conducted a comprehensive analysis to identify STAT3 3'UTR fragment utilized for its subcellular localization in cell protrusions and proved the direct binding between FMRP and STAT3 3'UTR by RIP. FMRP was knocked-down and mutated to further demonstrate its involvement in the translational regulation of STAT3. An orthotopic liver tumor model was employed to validate the interactions between FMRP/STAT3 mRNA contributed to the growth and metastasis of HCC. Lastly, clinical specimens of HCC were also studied to suggest the involvement of FMRP in cancer metastasis.

This is a comprehensive piece of study and the rationales were clear. On the large, the data is original and is sufficient to support the scientific conclusions.

Some minor suggestions listed below will help to improve the manuscript:

1. The quality of the image presented in Figure 3F was poor and out of focus, the protrusions could not be clearly identified.
2. Scale bars should be included in all images.
3. Try to provide consistent typeface and font size within a single figure.
4. The cartoon illustration provided in Fig.7 was not professionally done.
5. Introducing the biological function of FMRP in axons is insufficient. In this study, the authors showed that the expression of FMRP was strikingly high in HCC compared to normal tissue. Since STAT3 plays an oncogenic role in many other types of cancers, these results should also be discussed with previous studies or general knowledge of FMRP in cancers.

Reviewer #2 (Remarks to the Author):

Stat3 mRNA accumulates in cell protrusions of metastatic HCC cells. The authors claim that the FMRP protein binds the 3' UTR of Stat3 mRNA to help it localize in the protrusions.

The paper could have some potential, but there are serious flaws that make it unacceptable for publication.

1. It has been extensively documented that engagement of cadherins, as occurs in cultured cells, has a dramatic effect upon Stat3, ptyr705 phosphorylation and transcriptional activity. Since Stat3 activates its own promotor, an increase in Stat3-705 would increase total Stat3 protein as well, but to a lesser extent. This fact is not taken into account at all in this paper, and this unfortunately makes most of the cell culture results very hard to interpret. For example:
2. The diagram in Fig. 1A shows cells growing on transwell membranes with no contact with each other. If there was contact however, Stat3 activity might be much higher. If Stat3 promotes metastasis, did the cells have more protrusions when grown to high densities?
3. Similarly, the degree of downregulation of Stat3 must be tested at a range of densities. If an shRNA (or any other factor) simply slows down cell growth, then perhaps the cells would be less confluent and Stat3-705 levels much lower.
4. In Fig. 4A, it is stated in the legend and the text that Stat3 protein was increased upon IL6 treatment, which would be counter to the expected dogma. However the Figure shows that Stat3-p (presumably ptyr705) is increased, as expected, while total Stat3 is almost the same. The fact that mRNA levels were not affected is taken as an indication that IL6 enhances Stat3 translation. However, it could be Stat3 protein degradation, mRNA stability and other factors. This section is very confusing.
5. Further down in page 13 it is stated that in FMRP knockdown lines Stat3 was downregulated but the mRNA was unaffected. This was taken as an indication that FMRP might modulate Stat3 transcription.

If mRNA was unaffected, how can transcription be affected?

6. The next section deals with the effect of FMRP knockdown upon cell migration. Again, cell density was not taken into account. If FMRP knockdown slows down cell growth, then it is likely that the cells might be less dense, hence have much lower Stat3-p705 levels and this might affect their migration too. In any event, cell density must be taken into account by testing at a range of densities from ~50% to a few days post 100% confluence before any conclusions can be drawn.

7. At the bottom of page 9, it is stated that IHC staining showed that Stat3 was highly expressed in carcinoma tissues. What was the expression of ptyr705-Stat3?

Fig. 2A: The magnification is not high enough but in "Stat3 overall" (ie full-length Stat3) it looks like Stat3-FISH was up in the whole cells, not just the protrusions.

Minor points:

There are numerous grammatical and syntactical errors that make it sometimes hard to follow. By p-Stat3 the authors may mean ptyr705-Stat3, not p727.

Reviewer #3 (Remarks to the Author):

The manuscript "Involvement of FMRP in STAT3 mRNA localization to cellular protrusions and local translation promoting HCC metastasis" by Shen et al deals with the mechanism with which STAT3 mRNA is recruited at the cell protrusions and its functional role in invasiveness. The data presented here are very interest although some results are overinterpreted. Overall the results are very preliminary. The finding that FMRP could mediated cell proliferation, migration and invasion is not completely new being already known in other tumors as breast cancer (Cell Death Dis. 2017 Nov 16;8(11)) and melanoma (Cell Death Dis. 2017 Nov 16;8(11)). All in vitro experiments have been conducted in one cell line only.

Specific criticisms

Figure 1. The authors claim that in figure 1B is shown the STAT3 mRNA by RT-PCR assay. Unfortunately no RTPCR is shown.

Figure2B. The authors claim that the 3' UTR is necessary for the STAT3 mRNA localization at the protrusions. I do not completely agree with this interpretation. Indeed even the delta2 would seem to do so. The images of the cells are cut in such a way that the protrusions are not seen well. Quantification of this data is absolutely necessary just as it is necessary to show images with more cells and where protrusions are not cut out of the field.

Figure2G. As above in figure 2B also here is necessary to show images with more cells and where protrusions are not cut out of the field and quantification of this data should be presented.

Figure 3A,B. Rip should be repeated on isolated cell protrusions.

Figure3C. Also in this figure it is very difficult to see what the authors claim. How extended is the localization and where? in the cytoplasm? in the protrusions? No quantification data are presented FMRI binds the 3'UTR of STAT3. However there are no indications of which part of the 3'UTR is involved despite the fact that in the previous figure 3'UTR mutants have been used. Studies of colocalization between mutated STAT3 3'UTR and FMRI as well as gel shifts conducted with mutated probes would be added.

Figure3F. Quantification is mandatory.

Figure4C. The protein level of STAT3 was only slightly (not markedly as the authors claim) downregulated in FMRP knockdown cells. Moreover the fact that the mRNA level of STAT3 had no obvious change (Figure 4.D), indicates that FMRP do not (and not might also as the author claim) modulate the STAT3 expression at the transcriptional level.

The experiments (WB)in Figure 4D and H are not of sufficient quality. In particular in 4H different exposure time of the gel should be used and/or the experiments must be repeated to have a more

consistent loading control.
Giulia Piaggio

Reply to Comments

Dear Editor:

Thank you very much for your comments for our manuscript entitled “Involvement of FMRP in STAT3 mRNA localization to cellular protrusions and local translation promoting HCC metastasis”. The reviewers’ comments are very helpful for revising and improving our paper. We carefully revised our manuscript point-by-point according to the reviewers’ comments and highlighted the changes in the revised manuscript by using colored text. Please find the reply to comments as follows.

Reviewer #1

1) **Question:** *The quality of the image presented in Figure 3F was poor and out of focus, the protrusions could not be clearly identified.*

Answer: Thank you very much for your suggestion. According to your comment, we replaced the images of Figure 3.F with higher quality ones (The Figure 3.F has been re-organized as Figure 3.J in the revised manuscript). The cell protrusions are marked with white arrows.

2) **Question:** *Scale bars should be included in all images.*

Answer: Thank you very much for your comment. We have supplemented the scale bars in all needed images.

3) **Question:** *Try to provide consistent typeface and font size within a single figure.*

Answer: Thanks for your suggestion. According to your comment, we checked and provided consistent typeface and font size in every figure in the revised manuscript.

4) **Question:** *The cartoon illustration provided in Fig.7 was not professionally done.*

Answer: Thanks for the suggestion. According to reviewer’s comments, we deleted the Figure 7 in the revised manuscript.

5) **Question:** *Introducing the biological function of FMRP in axons is insufficient. In this study, the authors showed that the expression of FMRP was strikingly high in HCC compared to normal tissue. Since STAT3 plays an oncogenic role in many other types of cancers, these results should also be discussed with previous studies or general knowledge of FMRP in cancers.*

Answer: Thanks for the suggestion. We entirely agree with your point. According to your comments, we provided more rationales in the Background and Discussion sections for the current study. The changes were highlighted by using colored text in the revised manuscript (the newly added parts are marked yellow; you can read it together with the content marked in green).

Reviewer #2

1) **Question:** *It has been extensively documented that engagement of cadherins, as occurs in cultured cells, has a dramatic effect upon Stat3, ptyr705 phosphorylation and transcriptional activity. Since Stat3 activates its own promotor, an increase in Stat3-705 would increase total Stat3 protein as well, but to a lesser extent. This fact is not taken into account at all in this paper, and this unfortunately makes most of the cell culture results very hard to interpret. For example:*

Answer: Thank you very much for your comment. As you said, numerous studies have revealed the functional significance of cadherins in the regulation of STAT3, ptyr705 phosphorylation and transcriptional activity [1-5]. For example, the study uncovered that the cell density triggers a dramatic activation of STAT3, and the ablation of E-cadherin inhibits the cell density-mediated activation [3]. We evaluated the level of STAT3-pTyr705 in different cell densities and found that STAT3 can indeed be activated (Figure A, B). However, biological phenomena formed in cells are based on intricate signal networks, usually caused by a variety of reasons. Our research mainly focused on the regulation of STAT3 mRNA localization in protrusions in HCC metastasis, not the regulation of STAT3 activation. We used a same cell density for sampling and testing during experiments, and found that FMRP could promote metastasis by regulating the localization and translation of STAT3 mRNA.

This finding may be another molecular mechanism independent of cadherin, so the original manuscript did not pay much attention to the influence of cadherin in it. But we are glad to study the potential role of cadherin in the localization of mRNA in cell protrusions in the subsequent work. In addition, according to your comment, we re-designed the experiment by setting different cell densities and confirmed that the cell migration was notably suppressed by FMRP knockdown regardless of the cell density (please refer to the answer and figure of question 6).

A. HCCLM3 cells grown to the indicated confluences were photographed. **B.** Cell density upregulates the level of STAT3-pTyr705.

2) **Question:** *The diagram in Fig. 1A shows cells growing on transwell membranes with no contact with each other. If there was contact however, Stat3 activity might be much higher. If Stat3 promotes metastasis, did the cells have more protrusions when grown to high densities?*

Answer: Thanks for the suggestion. Sorry for that our diagram has misunderstood you. This schematic diagram is only for simple explanation of the experimental principle, and does not mean that there is no cellular contact. Generally, when the cell density reaches above 90%, we will start the experimental operation. Under such conditions, it is impossible to do without cellular contact.

3) **Question:** *Similarly, the degree of downregulation of Stat3 must be tested at a range of*

densities. If an shRNA (or any other factor) simply slows down cell growth, then perhaps the cells would be less confluent and Stat3-705 levels much lower.

Answer: Thank you very much for the suggestion. We can understand your concern. Studies have shown that STAT3 can be activated in a ligand-independent way by cell confluence in multiple cancer cell lines [1, 2]. But our purpose is to study the relationship between the mRNA location, translation and function of STAT3. So we evaluated the activation of STAT3 in a traditional way by using same cell density for sampling and testing during all experiments. Researches have revealed that STAT3 is able to promote metastasis of cancer cells [6-8]. Therefore, we only knockdown STAT3 by shRNA to verify this phenotype without set different cell densities in the manuscript.

4) **Question:** *In Fig. 4A, it is stated in the legend and the text that Stat3 protein was increased upon IL6 treatment, which would be counter to the expected dogma. However the Figure shows that Stat3-p (presumably ptyr705) is increased, as expected, while total Stat3 is almost the same. The fact that mRNA levels were not affected is taken as an indication that IL6 enhances Stat3 translation. However, it could be Stat3 protein degradation, mRNA stability and other factors. This section is very confusing.*

Answer: Thanks for the suggestion. We apologize for the confusion generated by the previous version of manuscript. According to your comments, we improved the experimental conditions (IL-6 treatment time) and repeated this series of experiments. The results are as follows: 1. the level of phosphorylated STAT3 was upregulated by IL-6 treatment (Figure 4.A). At the same conditions, nuclear phosphorylated STAT3 was also significantly increased (Figure 4.B), suggesting that IL-6 can be used as an activator of STAT3. This data was consistent with the expected dogma. 2. The protein level of STAT3 was increased after IL-6 treatment, while the mRNA level was unaffected, indicating that IL-6 might modulate STAT3 expression at a translational or post-translational level (Figure 4.A). 3. IF assay results uncovered that IL-6 notably enhanced the expression and subsequent nuclear importation of STAT3 protein (Figure 4.C). This data further confirms the above result1 and result2.

For your concern about result2, we separately used ActinomycinD (transcriptional inhibitor) and CHX (protein translational inhibitor) to treat cells and re-performed Q-PCR and western blot

assays. 1. The data showed that IL-6 was unable to affect the mRNA level of STAT3 with or without Actinomycin D treatment, excluding the possibility of IL-6 regulating the transcription or mRNA stability of STAT3 mRNA (Figure S2). 2. CHX treatment could abolish IL-6-induced increase of STAT3 protein, excluding the possibility of IL-6 regulating the protein stability of STAT3 (Figure 4.F). The above data revealed that IL-6 regulated STAT3 protein level at the translational level rather than post-translational level.

5) **Question:** *Further down in page 13 it is stated that in FMRP knockdown lines Stat3 was downregulated but the mRNA was unaffected. This was taken as an indication that FMRP might modulate Stat3 transcription. If mRNA was unaffected, how can transcription be affected?*

Answer: Thanks for the comment. We're so sorry for our fault by writing "translation" as "transcription". Our original intention is that FMRP might modulate the translation of STAT3. In the revised manuscript, we have corrected the error and highlighted the changes by using colored text.

6) **Question:** *The next section deals with the effect of FMRP knockdown upon cell migration. Again, cell density was not taken into account. If FMRP knockdown slows down cell growth, then it is likely that the cells might be less dense, hence have much lower Stat3-p705 levels and this might affect their migration too. In any event, cell density must be taken into account by testing at a range of densities from ~50% to a few days post 100% confluence before any conclusions can be drawn.*

Answer: Thank you very much for your suggestion. Literatures have reported that the difference of cell density can significantly affect cell migration [1-3]. Therefore, we re-designed the experiment according to reviewer's comment, to evaluate the effect of FMRP on cell migration. We used the HCCLM3-Con and HCCLM3-sh-FMRP#1 cells to perform the wound healing assay. Four cell densities (50%, 80%, 100% and 100% + 1 day) were selected for the wound healing assay (Figure A). Wound closure was determined after scratch for 72h. The result indicated that the higher the cell density, the faster the scratch healing. More importantly, at each cell density (except for 50%, this cell density is too low for HCCLM3 to

heal the wound in 72h), the cell migration was notably suppressed with FMRP knockdown, indicating that FMRP can indeed affect cell migration (Figure B and C).

A. HCCLM3 cells grown to the indicated confluences were photographed. **B and C.** Effect of FMRP knockdown on the rate of wound healing under different cell densities.

7-1) **Question:** At the bottom of page 9, it is stated that IHC staining showed that Stat3 was highly expressed in carcinoma tissues. What was the expression of ptyr705-Stat3?

Answer: Thanks for the suggestion. According to the comment, we originally planned to use

the same batch of samples in Figure 1.D to detect the level of p-STAT3. But the same batch of immunohistochemistry samples at that time has been used up. Only some tissue protein samples of the same batch are left. Therefore, we examined the expression level of p-STAT3 (ptyr705) in these samples by western blot assay. The result showed that the expression level of ptyr705-STAT3 was highly expressed in most HCC tissues (3/4) compared with that in non-carcinoma tissues (Figure A).

7-2) **Question:** *Fig. 2A: The magnification is not high enough but in “Stat3 overall” (ie full-length Stat3) it looks like Stat3-FISH was up in the whole cells, not just the protrusions.*

Answer: Thank you very much for the suggestion. Firstly, we modified the annotation errors in Figure 2.A and B, and replaced the figures in Figure 2.B with a low-power field of view, which can show more cell states. 1. We performed quantitative analysis in Figure 2.B, and the data showed that the abundances of protrusion-localization in STAT3 overall, delta2 and delta3 groups were significantly higher than that in the control and delta1 groups, indicating that STAT3 mRNA 3'UTR can indeed affect the protrusion-localization of STAT3 mRNA. 2. In response to the question you mentioned, we found that as you said, not only the STAT3 overall group, but also the delta2 and delta3 groups, the abundance of STAT3 mRNA has increased to a certain extent in the whole cell. This phenomenon may be due to the presence of 3'UTR, which is closely related to the regulation of mRNA stability [9, 10].

8) **Question:** *There are numerous grammatical and syntactical errors that make it sometimes hard to follow.*

Answer: Thanks for the comment. We apologize for the poor language of our manuscript. We worked on the manuscript for a long time and the repeated addition and removal of sentences obviously led to poor readability. According to your suggestion, we asked professional editing

service (American Journal Experts) to edit and polish the manuscript. We really hope that the levels of logic and language have been substantially improved. The associated certificate provided by American Journal Experts has been uploaded as a supplementary file.

9) **Question:** *By p-Stat3 the authors may mean ptyr705-Stat3, not p727.*

Answer: Thanks for the comment. As you said, we are referring to the ptyr705-Stat3.

Reviewer #3

1) **Question:** *Figure 1. The authors claim that in figure 1B is shown the STAT3 mRNA by RT-PCR assay. Unfortunately no RT-PCR is shown.*

Answer: Thanks for the comment. Sorry for the confusion caused by our mistake. The Figure 1.B showed the quality of the extracted protrusion RNA. 28sRNA and 18sRNA bands are clearer, and 5s at the band is not clear, suggesting that RNA is not contaminated with DNA. And the Figure S2 showed the STAT3 mRNA (STAT3 α) of cell protrusion examined by RT-PCR assay. We have corrected the above errors and swapped the positions of Figure S2 and Figure 1.B for description. The changes were highlighted in the revised manuscript by using colored text.

2) **Question:** *Figure2B. The authors claim that the 3' UTR is necessary for the STAT3 mRNA localization at the protrusions. I do not completely agree with this interpretation. Indeed even the delta2 would seem to do so. The images of the cells are cut in such a way that the protrusions are not seen well. Quantification of this data is absolutely necessary just as it is necessary to show images with more cells and where protrusions are not cut out of the field.*

Answer: Thanks for the comment. Sorry, we have mistaken the data in Figure 2.A and B. In the delta2 group, only the SH2 and TAD domains were deleted, and the 3'UTR of STAT3 mRNA was retained. This explained why the delta2 group also had the signal of protrusion-localization. According to your suggestion, firstly, we modified the errors and replaced the figures of Figure 2.B with a low-power field of view, which could show more cell states. Secondly, we performed quantitative analysis in Figure 2.B, and the data showed

that the abundances of protrusion-localization in STAT3 overall, delta2 and delta3 groups were significantly higher than that in the control and delta1 groups (termed as Figure 2.C in the revised manuscript), indicating that STAT3 mRNA 3'UTR can indeed affect the protrusion-localization of STAT3 mRNA.

3) **Question:** *Figure 2G. As above in figure 2B also here is necessary to show images with more cells and where protrusions are not cut out of the field and quantification of this data should be presented.*

Answer: Thanks for the comment. According to your suggestion, we replaced the figures in Figure 2.G with a low-power field of view, which can show more cell states (The Figure 2.G has been re-organized as Figure 2.H in the revised manuscript). The quantification analysis in Figure 2.G was also performed (termed as Figure 2.I in the revised manuscript). The changes were highlighted in the revised manuscript by using colored text.

4) **Question:** *Figure 3A,B. Rip should be repeated on isolated cell protrusions.*

Answer: Thanks for the comment. According to your suggestion, we repeated the RIP assay in the isolated cell protrusions of HCCLM3 cells. The results showed that FMRP interacted with STAT3 mRNA in cell protrusions (Figure 3.C). The changes were highlighted in the revised manuscript by using colored text.

5) **Question:** *Figure 3C. Also in this figure it is very difficult to see what the authors claim. How extended is the localization and where? in the cytoplasm? in the protrusions? No quantification data are presented.*

Answer: Thanks for the comment. According to your suggestion, we performed quantitative analysis of colocalization in Figure 3.C by image J software (The Figure 3.C has been re-organized as Figure 3.D in the revised manuscript). Firstly, we used the image calculator function to correct the background of the image. After unifying the background, the colocalization finder tool was used to analyze the Pearson correlation coefficient (The closer the Pearson coefficient is to 1, the higher the correlation) [11-13]. In addition, the co-localization image was converted into a visual scatter plot (Figure. A, termed as Figure 3.E

in the revised manuscript). The figure showed that most of the points are distributed on the diagonal, and the Pearson coefficient is 0.7374. In this case, it indicated that FMRP and STAT3 mRNA are co-localized in HCCLM3 cells. The data in Figure 3.D revealed they were mainly co-localized in the cytoplasm and protrusions of cells. In the revised manuscript, the corresponding details have been added to the sections of Results and Materials and methods, and marked with yellow.

A. Quantification analysis of colocalization of FMRP and STAT3 mRNA in HCCLM3 cells.

6) **Question:** *FMRI binds the 3'UTR of STAT3. However there are no indications of which part of the 3'UTR is involved despite the fact that in the previous figure 3'UTR mutants have been used. Studies of colocalization between mutated STAT3 3'UTR and FMRI as well as gel shifts conducted with mutated probes would be added.*

Answer: Thanks for the comment. According to your suggestion, we performed the gel-mobility shift assay by adding the unlabeled STAT3 3'UTR MUT-1 and MUT-2. The data showed that the addition of unlabeled STAT3 3'UTR MUT-1 was unable to abolish the interaction of FMRP with radiolabeled STAT3 3'UTR, while the addition of unlabeled STAT3 3'UTR MUT-2 could abolish it, suggesting that the MUT-1 cannot interact with FMRP. This data indicated the loop1 of STAT3 3'UTR was responsible for the interaction between FMRP and STAT3 3'UTR (this data termed as Figure 3.G in the revised manuscript). The changes were highlighted in the revised manuscript by using colored text.

7) **Question:** *Figure3F. Quantification is mandatory.*

Answer: Thanks for the comment. The quantification of fluorescence data in Figure 3.F was performed (The Figure 3.F has been re-organized as Figure 3.J in the revised manuscript, the right panel is the quantification data).

8) **Question:** *Figure 4C. The protein level of STAT3 was only slightly (not markedly as the authors claim) downregulated in FMRP knockdown cells. Moreover the fact that the mRNA level of STAT3 had no obvious change (Figure 4.D), indicates that FMRP do not (and not might also as the author claim) modulate the STAT3 expression at the transcriptional level.*

Answer: Thanks for the comment. We suppose you are referring to the problem of Figure 4.D. Firstly, according to your suggestion, we re-performed the experiment in Figure 4.D. The data showed that the protein level of STAT3 was decreased in the FMRP knockdown cells. Secondly, we are so sorry for our fault by writing “translation” as “transcription”. Our original intention is that FMRP might modulate the translation of STAT3. In the revised manuscript, we have corrected the error and highlighted the changes by using colored text.

9) **Question:** *The experiments (WB) in Figure 4D and H are not of sufficient quality. In particular in 4H different exposure time of the gel should be used and/or the experiments must be repeated to have a more consistent loading control.*

Answer: Thanks for the comment. According to your suggestion, we repeated the experiment in Figure 4.D and Figure 4.H and improved the quality (The Figure 4.H has been re-organized as Figure 4.G in the revised manuscript).

References

1. Steinman, R.A., et al., *Activation of Stat3 by cell confluence reveals negative regulation of Stat3 by cdk2*. *Oncogene*, 2003. **22**(23): p. 3608-15.
2. Vultur, A., et al., *Cell-to-cell adhesion modulates Stat3 activity in normal and breast carcinoma cells*. *Oncogene*, 2004. **23**(15): p. 2600-16.
3. Arulanandam, R., et al., *Cadherin-cadherin engagement promotes cell survival via Rac1/Cdc42 and signal transducer and activator of transcription-3*. *Mol Cancer Res*, 2009. **7**(8): p. 1310-27.
4. Raptis, L., et al., *Beyond structure, to survival: activation of Stat3 by cadherin engagement*. *Biochem Cell Biol*, 2009. **87**(6): p. 835-43.
5. Geletu, M., et al., *Reciprocal regulation of the Cadherin-11/Stat3 axis by caveolin-1 in mouse fibroblasts and lung carcinoma cells*. *Biochim Biophys Acta Mol Cell Res*, 2018. **1865**(5): p. 794-802.
6. Zhu, H., et al., *AKR1C1 Activates STAT3 to Promote the Metastasis of Non-Small Cell Lung Cancer*. *Theranostics*, 2018. **8**(3): p. 676-692.
7. Zhang, X.P., et al., *PRMT1 Promoted HCC Growth and Metastasis In Vitro and In Vivo via Activating the STAT3 Signalling Pathway*. *Cell Physiol Biochem*, 2018. **47**(4): p. 1643-1654.
8. Lee, C. and S.T. Cheung, *STAT3: An Emerging Therapeutic Target for Hepatocellular Carcinoma*. *Cancers (Basel)*, 2019. **11**(11).
9. Misquitta, C.M., et al., *The role of 3'-untranslated region (3'-UTR) mediated mRNA stability in cardiovascular pathophysiology*. *Mol Cell Biochem*, 2001. **224**(1-2): p. 53-67.
10. Thiele, A., et al., *AU-rich elements and alternative splicing in the beta-catenin 3'UTR can influence the human beta-catenin mRNA stability*. *Exp Cell Res*, 2006. **312**(12): p. 2367-78.
11. Adler, J. and I. Parmryd, *Quantifying colocalization by correlation: the Pearson correlation coefficient is superior to the Mander's overlap coefficient*. *Cytometry A*, 2010. **77**(8): p. 733-42.
12. Bolte, S. and F.P. Cordelières, *A guided tour into subcellular colocalization analysis in light microscopy*. *J Microsc*, 2006. **224**(Pt 3): p. 213-32.
13. Zinchuk, V. and O. Grossenbacher-Zinchuk, *Recent advances in quantitative colocalization analysis: focus on neuroscience*. *Prog Histochem Cytochem*, 2009. **44**(3): p. 125-72.

REVIEWERS' COMMENTS:

Reviewer #2 (Remarks to the Author):

The revised manuscript addresses my concerns.

However, I would like to see the new data shown in the cover letter, incorporated into the text, or at least as Supplementary data.

The cell densities (~90%) must be mentioned clearly in the text.

Then the manuscript would be acceptable for publication.

Reviewer #3 (Remarks to the Author):

The authors addressed all criticisms. The manuscript is now acceptable.

Reply to Comments

Dear Editor:

Thank you very much for your comments for our manuscript entitled “FMRP regulates STAT3 mRNA localization to cellular protrusions and local translation to promote HCC metastasis”. The reviewers’ comments are very helpful for improving our paper. We carefully revised our manuscript according to the reviewers’ comments. Please find the reply to comments as follows.

Reviewer #2

1) **Question:** *The revised manuscript addresses my concerns. However, I would like to see the new data shown in the cover letter, incorporated into the text, or at least as Supplementary data. The cell densities (~90%) must be mentioned clearly in the text.*

Answer: Thank you very much for your suggestion. According to your comment, we incorporated the new data into the supplementary data (supplementary information) and clearly described it in the text.